# Mental Health and Adherence to COVID-19 Protective Behaviors among Cancer Patients during the COVID-19 Pandemic: An International, Multinational Cross-Sectional Study

**DOI:** 10.3390/cancers13246294

**Published:** 2021-12-15

**Authors:** Angelos P. Kassianos, Alexandros Georgiou, Maria Kyprianidou, Demetris Lamnisos, Jeļena Ļubenko, Giovambattista Presti, Valeria Squatrito, Marios Constantinou, Christiana Nicolaou, Savvas Papacostas, Gökçen Aydin, Yuen Yu Chong, Wai Tong Chien, Ho Yu Cheng, Francisco J. Ruiz, Maria B. Garcia-Martin, Diana Obando, Miguel A. Segura-Vargas, Vasilis S. Vasiliou, Louise McHugh, Stefan Höfer, Adriana Baban, David Dias Neto, Ana Nunes da Silva, Jean-Louis Monestès, Javier Alvarez-Galvez, Marisa Paez Blarrina, Francisco Montesinos, Sonsoles Valdivia Salas, Dorottya Őri, Bartosz Kleszcz, Raimo Lappalainen, Iva Ivanović, David Gosar, Frederick Dionne, Rhonda M. Merwin, Andreas Chatzittofis, Evangelia Konstantinou, Sofia Economidou, Andrew T. Gloster, Maria Karekla, Anastasia Constantinidou

**Affiliations:** 1Department of Psychology, University of Cyprus, Nicosia 2109, Cyprus; angelos.kassianos@ucl.ac.uk (A.P.K.); mkypri11@ucy.ac.cy (M.K.); 2Department of Applied Health Research, University College London, London WC1E 6BT, UK; 3Bank of Cyprus Oncology Centre, Nicosia 2006, Cyprus; Alexandros.Georgiou@bococ.org.cy (A.G.); evangelia.konstantinou@nhs.net (E.K.); noemails@hotmail.com (S.E.); 4Department of Health Sciences, European University of Cyprus, Nicosia 2109, Cyprus; D.Lamnisos@euc.ac.cy; 5Psychological Laboratory, Faculty of Public Health and Social Welfare, Riga Stradins University, LV-1007 Riga, Latvia; jelena.lubenko@rsu.lv; 6Human and Social Sciences Department, University of Enna Kore, 94100 Enna, Italy; giovambattista.presti@unikore.it (G.P.); valeria.squatrito@unikore.it (V.S.); 7Department of Psychology, University of Nicosia, Nicosia 1516, Cyprus; CONSTANTINOU.M@UNIC.AC.CY; 8Department of Nursing, School of Health Sciences, Cyprus University of Technology, Limassol 3041, Cyprus; c.nicolaou@cut.ac.cy; 9Cyprus Institute of Neurology and Genetics, Nicosia 2371, Cyprus; savvas@cing.ac.cy; 10Department of Psychological Counseling and Guidance, Hasan Kalyoncu University, Gaziantep 27010, Turkey; gokcen.aydin@hku.edu.tr; 11The Nethersole School of Nursing, Faculty of Medicine, The Chinese University of Hong Kong, Hong Kong SAR, China; conniechong@cuhk.edu.hk (Y.Y.C.); wtchien@cuhk.edu.hk (W.T.C.); hycheng@cuhk.edu.hk (H.Y.C.); 12Department of Psychology, Universidad de La Sabana, Chia 140013, Colombia; fransisco.ruizj@konradlorenz.edu.co (F.J.R.); maria.garcia27@unisabana.edu.co (M.B.G.-M.); diana.obando@unisabana.edu.co (D.O.); mseguravargas@unisabana.edu.co (M.A.S.-V.); 13School of Applied Psychology, University College Cork (UCC), T23 XE10 Cork, Ireland; v.vasiliou@ucc.ie; 14School of Psychology, University College Dublin, D04 V1W8 Dublin 4, Ireland; louise.mchugh@ucd.ie; 15Department of Medical Psychology, Medical University Innsbruck, 6020 Innsbruck, Austria; stefan.hoefer@i-med.ac.at; 16Department of Psychology, Babes-Bolyai University, 400015 Cluj-Napoca, Romania; adrianababan@psychology.ro; 17ISPA—Instituto Universitário, 1149-041 Lisboa, Portugal; d.neto@campus.ul.pt; 18Applied Psychology Research Center Capabilities & Inclusion, 1149-041 Lisbon, Portugal; 19Department of Psychology, University of Lisbon, 1649-013 Lisbon, Portugal; acsilva@psicologia.ulisboa.pt; 20LIP/PC2S, Université Grenoble Alpes, 38000 Grenoble, France; jean-louis.monestes@univ-grenoble-alpes.fr; 21Department of Biomedicine, Biotechnology and Public Health, University of Cadiz, 11009 Cadiz, Spain; javier.alvarezgalvez@uca.es; 22Instituto ACT, 28049 Madrid, Spain; marisa.paez@institutoact.es (M.P.B.); fransisco.montestinos@universidadeuropea.es (F.M.); 23Department of Psychology, Universidad Europea de Madrid, 28049 Madrid, Spain; 24Department of Psychology and Sociology, Universidad de Zaragoza, 50013 Zaragoza, Spain; sonsoval@unizar.es; 25Department of Mental Health, Heim Pal National Pediatric Institute, H-1089 Budapest, Hungary; oridorottya@heimpalkorhaz.hu; 26Institute of Behavioral Sciences, Semmelweis University, H-1089 Budapest, Hungary; 27Private Practice, 41-200 Sosnowiecz, Poland; bkleszcz@gmail.com; 28Department of Psychology, University of Jyväskylä, 40014 Jyväskylä, Finland; raimo.i.lappalainen@jyu.fi; 29Department of Child Psychiatry, Institute for Children’s Diseases, Clinical Centre of Montenegro, 81000 Podgorica, Montenegro; iva.ivanovic@kccg.me; 30Department of Child, Adolescent and Developmental Neurology, University Children’s Hospital Ljubljana, University Medical Center Ljubljana, 1000 Ljubljana, Slovenia; david.gosar@kclj.si; 31Département de Psychologie, Université du Québec à Trois-Rivières, Trois-Rivières, QC G8Z 4M3, Canada; frederick.dionne@uqtr.ca; 32Department of Psychiatry and Behavioral Science, Duke University, Durham, NC 27708, USA; rhonda.merwin@duke.edu; 33Medical School, University of Cyprus, Nicosia 1678, Cyprus; Chatzittofis.andreas@ucy.ac.cy; 34Division of Clinical Psychology & Intervention Science, Department of Psychology, University of Basel, 4055 Basel, Switzerland; andrew.gloster@unibas.ch; 35Cyprus Cancer Research Institute, Nicosia 2109, Cyprus

**Keywords:** cancer patients, mental health, health behaviors, protection behaviors, COVID-19

## Abstract

**Simple Summary:**

Whilst information on the impact of COVID-19 on cancer care continues to increase exponentially, little is known about the impact of COVID-19 on the mental health and coping behaviors of cancer patients. This study constitutes a sub-study of a large international survey conducted during the first wave of the pandemic, looking specifically at the impact of COVID-19 on the mental health and protective behaviors of cancer participants, compared to non-cancer participants. It also explored whether cancer participants perceived COVID-19 as a bigger threat compared to their cancer and whether this perception affected their psychological outcomes, such as their perceived level of stress. Overall, cancer participants appeared better adapted compared to non-cancer participants, well-functioning, resilient and able to adjust, and prepared to deal with what is otherwise a worldwide crisis; perhaps as a result of their previous cancer experience. Whilst good news, these results should not lead to a dismissal of the specific needs of cancer patients and as the pandemic drags on, cancer population dedicated studies should be performed to ensure adequate care for these patients.

**Abstract:**

A population-based cross-sectional study was conducted during the first COVID-19 wave, to examine the impact of COVID-19 on mental health using an anonymous online survey, enrolling 9565 individuals in 78 countries. The current sub-study examined the impact of the pandemic and the associated lockdown measures on the mental health, and protective behaviors of cancer patients in comparison to non-cancer participants. Furthermore, 264 participants from 30 different countries reported being cancer patients. The median age was 51.5 years, 79.9% were female, and 28% had breast cancer. Cancer participants reported higher self-efficacy to follow recommended national guidelines regarding COVID-19 protective behaviors compared to non-cancer participants (*p* < 0.01). They were less stressed (*p* < 0.01), more psychologically flexible (*p* < 0.01), and had higher levels of positive affect compared to non-cancer participants. Amongst cancer participants, the majority (80.3%) reported COVID-19, not their cancer, as their priority during the first wave of the pandemic and females reported higher levels of stress compared to males. In conclusion, cancer participants appeared to have handled the unpredictable nature of the first wave of the pandemic efficiently, with a positive attitude towards an unknown and otherwise frightening situation. Larger, cancer population specific and longitudinal studies are warranted to ensure adequate medical and psychological care for cancer patients.

## 1. Introduction

The unparalleled health crisis caused by COVID-19 has created a major impact on cancer care worldwide. Early reports suggested that the prevalence of cancer in patients with COVID-19 is high and that cancer patients and cancer survivors are at-risk population groups for COVID-19 [1,2]. More importantly, cancer patients appear to be at higher risk of developing severe complications associated with COVID 19 [1,3] leading to fatality rates of 10–30% as shown in single center studies [3,4] and a mortality rate of 25.6% in a pooled analysis of 52 studies [5]. Cancer-specific characteristics including cancer type—hematological malignancies, lung cancer—and metastatic (versus early) disease, but also other risk factors which may coexist with cancer including older age and multiple comorbidities, have rendered patients with cancer more vulnerable to SARS–COV-2 [4,6,7,8,9] and its sequalae. Therefore, understanding how cancer patients adhere to protective behaviors, recommended by local and international organizations, is important.

The Health Belief Model [10] is one of the theories used, to explain adherence to behaviors which can pertain to how vulnerable individuals consider themselves regarding a disease (perceived susceptibility), and how severe they consider the disease (perceived severity). The model has been used to understand COVID-19 related behaviors, such as intentions to vaccinate [11], adopting contact tracing [12], and adherence to protective behaviors [13]; but how this evidence translates to cancer patients’ behaviors is yet to be examined. There can be differences in how cancer patients (similarly to other chronic patients) adhere to self-protective behaviors reflecting their response to adjustments in risk assessment found elsewhere [14]. Thus, an individual’s assessment of risk may influence their behavior [15].

Many studies focused on the role of anticancer treatment particularly systemic therapy during the pandemic, with some but not all, reporting worsening of COVID-19-related outcomes, essentially increased mortality, in cancer patients undergoing active therapy [16,17]. These results prompted the implementation of updated recommendations by professional oncology societies on most cancer types to aid treatment decision making during the pandemic [18,19,20,21,22,23,24,25]. Yet, the challenges in cancer care linked directly or indirectly to COVID-19 are multifaceted. Delayed diagnosis, disruption of therapies, interruption or downscaling of screening programs, and follow-ups have been documented since the start of the pandemic in March 2020 [26]. Although long-term consequences may be indecipherable at present, predictive modelling indicated increases in the number of deaths due to delays in cancer diagnosis [27] and a detrimental impact on healthcare finances due to delayed access to cancer services [28].

Whilst information on all aforementioned COVID-19-influenced areas of cancer care continues to build exponentially, little is known about the impact of COVID-19 on cancer patients’ mental health. Since the COVID-19 pandemic, cancer patients and survivors have suffered from financial strains [29], especially for those with low income [30]; increased levels of loneliness [31] and their health-related quality of life, mainly their social, emotional, and cognitive functioning [32]. At a general population level, the abrupt and profound disruption of daily routines, has unsurprisingly brought increases in psychological distress, higher anxiety and depression levels, anger, and confusion [33,34,35]. The psychological implications on wellbeing have been explored in different countries [36,37,38,39,40,41,42], different professional groups including medical professionals and nurses [43,44,45], and in different population groups [46,47,48]. At the same time, identifying factors that have demonstrated facilitating wellbeing, such as psychological flexibility (ability to cope with, accept, and adjust to difficult situations) and mindfulness qualities (ability to be in the present moment rather than lost in past or future thinking) [49,50,51,52], social support [53,54], and tackling the impact of income loss, [55] can help to develop interventions that can ameliorate the impact of stress.

An international population-based cross-sectional study was conducted during the first lockdown period (April 2020–June 2020) to explore how people across the world reacted to COVID-19 by examining outcomes of stress, depression, affect, and wellbeing [33]. With the participation of 9565 people from 78 countries, the results showed that on average about 10% of the sample had low levels of mental health whilst 50% had only moderate mental health. In the current study we focused on the responses of the cancer participants who participated in the international study.

In this study, we sought to examine: (1) the level of adherence to protective behaviors of cancer participants in comparison to non-cancer participants and whether the factors associated with adherence differ between cancer participants and those without cancer (protective behaviors hypothesis); (2) the impact of the pandemic and the associated lockdown measures on mental health and emotional status of cancer participants in comparison to non-cancer participants and whether the factors associated with elevated levels of stress differ between cancer participants and those without cancer (mental health and coping hypothesis); and (3) whether cancer participants perceived COVID-19 as a bigger threat compared to their cancer and whether perceiving COVID-19 as a bigger threat affected their psychological outcomes (beliefs about COVID-19 hypothesis). We hypothesized that (1) cancer participants being more susceptible to COVID-19 will demonstrate higher adherence to protective behaviors; (2) the mental health of cancer participants would be more highly affected compared to non-cancer participants. The third objective is exploratory and thus no a priori hypothesis was made.

## 2. Materials and Methods

Ethics approval was obtained from the Cyprus National Bioethics Committee (ref.: EEBK EP2020.01.60) followed by site approvals in participating countries. All participants provided electronic informed consent prior to completing the online survey (by clicking ‘yes’).

### 2.1. Participants

Individuals aged 18 and older, residing in any country were able to read in a study language (English, Greek, German, French, Spanish, Turkish, Dutch, Latvian, Italian, Portuguese, Finnish, Slovenian, Polish, Romanian, Hong Kong, Hungarian, Montenegrin, Persian) were eligible to participate. There were no other exclusion criteria. Participants who agreed to participate in the study were invited to provide electronic informed consent and completed a 20-min online survey via a secured Google platform. No incentives were provided for participation. Participants were able to enroll in the study only once.

### 2.2. Study Design

A population-based cross-sectional study was conducted using an anonymous online survey which was distributed directly by the thirty-three participating universities to students and academic staff and through respective websites. The survey was also distributed through local press, social media, professional networks, local hospitals and health centers, social institutions, and through professional groups’ email lists in the participating countries. Data were collected for a period of two months—between 7 April and 7 June 2020—during which time a state of emergency for COVID-19 was declared by the World Health Organization (WHO) as well as in the majority of countries participating in the study.

### 2.3. Study Measures

The participants were assessed with well validated and established measures and in cases where measures were not already available in a language, they were subject to forward and backward translation procedures [56]. All measures were selected after a consensus agreement among the research members of the study.

#### 2.3.1. Measures for All Participants

All participants responded to a number of measures which were subsequently used for comparisons between cancer participants and non-cancer participants.

Socio-demographic and medical characteristics—including age, gender, marital status, employment status, educational level, parenthood status and living conditions, as well as cancer site at diagnosis were documented.

Lockdown and COVID-19 infection information—including length of lockdown, whether participants needed to leave home for work, had any change in their finances, whether they were able to obtain basic supplies, and the amount of living space they were confined in during the lockdown. Participants were also asked whether they, their partner, or significant other was diagnosed with COVID-19 at the time of their participation.

Adherence to COVID-19 protective behaviors (isolating, keeping distance, and hand washing), self-efficacy, and intentions to follow protective behaviors were also asked. Details on the questions asked on these measures are summarized in Appendix A. In general, participants were asked to respond on a Likert-type scale for each behavior ranging from 0 to 10 and one question on whether they intended to follow recommended behaviors for the following week with a 7-point Likert scale. Self-efficacy was assessed using an adapted version of the new general self-efficacy scale [57] with a computed score for each participant with higher scores indicating higher self-efficacy.

Social support and family functioning—Participants responded to Oslo Social Support Scale (OSS), a well-validated instrument assessing social determinants of health [58], and the Brief Assessment of Family Functioning (BAFFS), a measure with adequate internal consistency, construct, and concurrent validity assessing family functioning [59]

Mental Health—Participants responded to questions related to perceived stress (Perceived Stress Scale, PSS) [60], depressive symptomatology (Multidimensional State Boredom Scale, MSBS) [61], wellbeing (Mental Health Continuum Short Form for Adults, MHCSF) [62], positive/negative affect (Positive and Negative Affect Scale, PANAS) [63], mindfulness (Cognitive Affective Mindfulness Scale, CAMS) [64], and psychological flexibility (Psy-Flex) [65]. These measures’ factorial, convergent, content, and construct validity as well as reliability has been well established with cancer patients and the general population in the literature [66,67,68,69]. Details of mental health parameters and the corresponding measures are available in Appendix A and scoring information are available elsewhere [33].

Beliefs about COVID-19—A modified version of a questionnaire that measures the Health Beliefs Model’s perceived susceptibility and perceived severity parameters was used [70].

#### 2.3.2. Measures for Cancer Participants Only

Apart from the backbone questionnaire completed by all participants, specific questions were addressed to cancer participants only.

Cancer information—Participants reported the type of cancer, whether they were on anti-cancer therapy at the time of the survey, and whether the therapy was adjuvant (preventative) or for active disease.

Feelings towards cancer and COVID-19—Participants responded to three questions assessing what scared them the most at that moment, what they thought could harm them more, and what was their health priority at that moment (cancer or COVID-19). Subsequently their response to the question about priority, was used as an outcome in logistic regression models.

### 2.4. Statistical Analyses

Participants’ characteristics are presented descriptively using mean ± standard deviation (SD) for continuous variables with normal distributions and as median and interquartile range (IQR) for measures that do not follow the normal distribution.

For the beliefs about COVID-19 hypothesis and the protective behaviors hypothesis, Pearson’s chi-square test was employed to detect any differences between the variable on cancer diagnosis (cancer participants vs. those without) with the variable on their current priority (cancer vs. COVID-19). *T*-test and Kolmogorov–Smirnov test were applied to detect any differences between cancer participants and non-cancer participants and those with cancer vs. COVID-19 as their priority, and the continuous characteristics of the participants. The following cut-off values were used for the evaluation of the effect sizes: ‘tiny’ ≤ 0.05, ‘very small’ from 0.05 to 0.10, ‘small’ from 0.10 to 0.20, ‘medium’ from 0.20 to 0.30, ‘large’ from 0.30 to 0.40 and ‘very large’ > 0.40 [71].

For the protective behaviors hypothesis and the mental health and coping hypothesis, multivariate linear regression modelling was performed to evaluate the significance of different psychological and behavioral parameters on stress and in the level of keeping distance which constitutes behavior that was newly introduced into people’s routine [72] after accounting for different demographic and socio-economic characteristics, and psychological flexibility. Specifically, multivariate linear regression models were used, adjusting for different socio-demographic and socio-economic characteristics (i.e., age, gender). Firstly, we added perceived social support (model 1), then perceived susceptibility and perceived severity of COVID-19 (model 2), change in finances during lockdown and availability of obtaining all the basic supplies (model 3), and psychological flexibility (model 4). Finally, we applied a multivariate regression model adjusted for all predictors including cancer as priority, age, gender, cancer type, whether they were receiving anti-cancer therapy, perceived social support, perceived susceptibility and perceived severity, change in finances during quarantine and availability of obtaining all the basic supplies, and psychological flexibility (model 5). Β-coefficients and the corresponding 95% CIs were reported. The statistical hypotheses were two-sided with statistical significance level set at α = 0.05. Statistical analysis was conducted using STATA 14.0 statistical software (Stata Corp, College Station, TX, USA).

## 3. Results

### 3.1. Participants

Of the 9565 participants, 264 reported being cancer patients (Table 1). The median age of cancer participants was 51.5 years whereas for the non-cancer participants was 34 years (*n* = 9301, *p* < 0.01). The majority of cancer participants were female (79.9%) with males generally being older (M = 55, IQR = 40, 64.5) than females (M = 50, IQR = 40, 59, *p* = 0.04) and reporting worse losses in terms of their personal finances (44% vs. 25%, *p* = 0.02). Female cancer patients reported higher levels of stress (median = 15, IQR = 10, 21) than men (median = 12, IQR = 9, 17, *p* = 0.03). Cancer participants were residents in 30 different countries (see Appendix A). The majority were women with breast cancer (28%), followed by women with cancers of the female reproductive system (22%). Almost one third (31%) reported receiving adjuvant (preventative) therapy at the time of the survey, whilst 12% were receiving therapy for active disease. The remaining 57% were not receiving any therapy at the time. Most participants were working full time with more cancer participants being married (53%) compared to those without cancer (36%, *p* < 0.01). All information on participants’ sociodemographic information and further comparisons between cancer and non-cancer participants, as well as male and female cancer participants, are presented in Table 1, Table 2, Appendix A, and Appendix A, respectively.

### 3.2. Comparisons between Cancer Participants and Participants Not Reporting Cancer

#### 3.2.1. COVID-19 Protective Behaviors

Participants reporting cancer diagnosis were more likely to keep the recommended physical distance from other people (median = 10, IQR = 9, 10) when going out compared to non-cancer participants with a small effect size (median = 9, IQR = 8, 10, *p* < 0.01, d = 0.19). There were no differences between the two groups on limiting unnecessary traveling and washing hands (Table 2). Moreover, cancer participants reported a slightly higher level of self-efficacy on following the recommended national guidelines on COVID-19 protective behaviors compared to non-cancer participants with a small effect size (*p* < 0.01, d = 0.2). In addition, a higher percentage of cancer participants (67%) stated that they intended on following the recommendations for social distancing for the following week compared to non-cancer participants (57%, *p* < 0.01).

#### 3.2.2. Coping Styles, Social Support and Family Functioning

There were no significant differences between the two groups on coping with stressors including using emotional support, humor, and self-blaming (Appendix A). A higher number of cancer participants reported more social support (31%) compared to non-cancer participants (23%) but with tiny effect size (*p* < 0.01, d = 0.03). There were no significant differences between the two groups on family functioning (Table 2).

#### 3.2.3. Mental Health Outcomes

Cancer participants were less stressed (median = 15, IQR = 10, 20) than non-cancer participants with a medium effect size (median = 17, IQR = 12, 22, *p* < 0.01, d = 0.27). There were some differences between the two groups in terms of the three dimensions of wellbeing with largest differences reported on eudemonic psychological wellbeing with cancer participants (median = 23, IQR = 18, 26) scoring higher than non-cancer participants with medium effect size (median = 21, IQR = 15, 24, *p* < 0.01, d = 0.24). For all dimensions, cancer participants reported higher levels of wellbeing with one in two flourishing (50%) and 41.9% having moderate status of mental health (Table 2). Similar trends were found for non-cancer participants but with fewer of them flourishing (39.6%) compared to cancer patients. Cancer participants reported slightly higher levels of positive affect (median = 30, IQR = 25, 37) and thus propensity to experience positive emotions and interact with others positively compared to non-cancer participants with moderate effect size (median = 29, IQR = 23, 35, *p* < 0.01, d = 0.21). Further, cancer participants reported lower levels of negative affect (median = 26, IQR = 20, 33) and thus less tendency to experience the world in a negative way, compared to non-cancer participants with small effect size (median = 28, IQR = 25, 38, *p* < 0.01, d = 0.19). Finally, cancer participants were more psychologically flexible and thus taking their thoughts lightly, accepting their experiences, and engaging in what is important to them in the face of challenging situations compared to non-cancer participants, with large effect size (*p* < 0.01, d = 0.37) (Table 2). Furthermore, cancer participants reported being more mindful (living more in the here and now) compared to non-cancer participants (*p* = 0.04, d = 0.17) (Table 2).

#### 3.2.4. Beliefs towards COVID-19

Cancer participants felt more susceptible to being infected with COVID-19 (median = 11, IQR = 7.5, 14) compared to non-cancer participants with very large effect size (median = 8, IQR = 6, 11, *p* < 0.01, d = 0.56) and considered the disease to be slightly more severe (median = 14, IQR = 12, 17) compared to non-cancer participants with large effect size (median = 13, IQR = 10, 15, *p* < 0.01, d = 0.37).

### 3.3. Predictors of Stress and of Adhering to COVID-19 Protective Behaviors

Among cancer participants, in the multivariate model adjusted for all predictors (social support, perceived susceptibility, perceived change in finances during quarantine, availability of obtaining all the basic supplies, and psychological flexibility), results suggested that it was more likely to adhere to keeping physical distances when going out when feeling less susceptible to COVID-19, perceiving the disease as more severe, having losses in finances due to the pandemic, and being psychologically flexible (model 5, Table 3). In the second multivariate model adjusted for all predictors, we found females reporting higher levels of stress compared to males. Higher levels of stress were also related to increased perceived susceptibility to COVID-19, not being able to obtain all the basic supplies, and being less psychologically flexible (model 5, Table 4).

### 3.4. Comparisons according to Participants’ Primary Health Concern (Priority): Cancer or COVID-19

We investigated cancer participants further according to whether they reported having cancer as their primary health concern (cancer priority group) or COVID-19 (COVID-19 priority group) during the study period. An impressive 80.3% of cancer participants reported that COVID-19 was their top priority at the time compared to the remaining 19.7% who continued to view cancer as their top priority. Those in the cancer priority group were less likely to be university students (2%) than those in the COVID-19 priority group (12%, *p* = 0.04) but they did not significantly differ in other socio-demographic characteristics. All characteristics of both groups are available in Table 5 and in Appendix A.

#### 3.4.1. COVID-19 Protective Behaviors

There were no statistically significant differences between the cancer priority group (56%) or the COVID-19 priority group (73%) on COVID-19 protective behaviors and self-efficacy. However, it was more likely for those in the COVID-19 priority group (78%) to report that following the recommended national guidelines was important to them, compared to those in the cancer priority group (Table 5). Specifically, 62.1% of the COVID-19 priority group and 51.9% of the cancer priority group reported keeping distance from other people when going out all the time, respectively. Moreover, among COVID-19 priority group, 73.2% reported self-isolating and limiting unnecessary travelling according to national guidelines while the corresponding percentage among the cancer priority group was 55.8%. Finally, a similar percentage of individuals among the cancer priority group (65.3%) and the COVID-19 priority group (61.5%) reported washing their hands regularly with water and soap. Participants who had COVID-19 as their priority considered the disease as more severe (median = 15, IQR = 13, 17) compared to those that had cancer as their priority, with very large effect size (median = 12.5, IQR = 9, 15, *p* < 0.01, d = 0.74) (Table 5).

#### 3.4.2. Mental Health Outcomes

There were no significant differences between the two groups on their perceived stress, mindfulness, pro-social behavior, boredom, mental wellbeing, and psychological flexibility (Appendix A). Those in the cancer priority group reported higher positive affect (median = 32, IQR = 28.5, 38) compared to those in the COVID-19 priority group with large effect size (median = 30, IQR = 24, 37, *p* < 0.03, d = 0.34) suggesting they were more likely to experience positive emotions and interact with others positively despite challenges in their lives (Table 5). No differences were found between the two groups on negative affect.

## 4. Discussion

The implications of the COVID-19 pandemic on cancer patients and cancer care are feared to be immense. This study focused on the impact of COVID-19 on the mental health of cancer participants and their adherence to COVID-19 protective behaviors through an international population-based cross-sectional study using an anonymous online questionnaire. The analyses included comparisons of the parameters examined in the primary study [33] between cancer participants and non-cancer participants, aiming at describing the characteristics of the cancer subgroup and examining whether cancer participants’ behaviors, background experience with a chronic disease, and coping mechanisms led to different responses compared to those of non-cancer participants as well as comparing cancer participants according to their current health priority (cancer or COVID-19).

Examining the characteristics of the group of participants who reported being cancer patients, it primarily comprised of women with either breast cancer or cancer of the female reproductive system, however patients with different cancer types, from different age groups and different countries, participated in the study. About one third (31%) of the participants were receiving adjuvant treatment, 12% were receiving treatment for active disease, but the majority were not receiving any treatment at the time of the study, probably representing either cancer survivors, patients whose disease was under control, or both. In line with other studies, women with cancer were more likely to experience higher levels of stress compared to men [73].

### 4.1. Adherence to COVID-19 Protective Behaviors

In line with our first hypothesis, cancer participants demonstrated higher adherence to protective behaviors compared to non-cancer participants. It is important to note that during the data collection no vaccine was available for COVID-19. This may have affected study findings, since individual’s perceived susceptibility to COVID-19 may be affected and this can impact behavior. Cancer participants reported being more likely to keep physical distancing when going out, had higher self-efficacy to follow recommended protective behaviors, and stronger intentions to do so than those without cancer. This may be attributed to higher levels of mortality and morbidity identified in chronic patients who consequently are more protective of themselves and others [74,75]. We further identified, that whether those with cancer adhered to protective behaviors or not was associated with their beliefs towards COVID-19 (susceptibility to COVID-19, considering it more severe), having lost finances, being more stressed, and generally being more psychologically flexible, thus more likely to adapt to changing life situations. Cancer participants compared to non-cancer participants who worried about COVID-19 were more likely to take protective measures against COVID-19, such as limiting unnecessary travelling. Therefore, beyond being diagnosed with cancer, their beliefs towards the disease were strongly associated with their behavior; in line with the Health Belief Model. No other statistically significant differences were identified in terms of socio-demographic characteristics, lockdown information, health behaviors, or social support factors between cancer participants who had COVID-19 as their priority during the first wave and those who did not.

Despite the fear of COVID-19 contamination, cancer participants presented with protective factors of high self-efficacy and an intention to follow recommended guidelines not only of social distancing but also of adhering to measures such as hand washing. This may be due to the general attitude of individuals who have learned to live with a chronic disease and demonstrate flexibility and adjustability in order to cope with the challenges and unpredictability of an underlying health condition. On the other hand, this may be associated with the fear they experienced with this new thread which appeared more severe than cancer itself and led the vast majority of patients to set COVID-19 as their main priority [76]. Furthermore, in all risk efficacy parameters, cancer participants scored higher than non-cancer participants, demonstrating their perception that they can deal with such a difficult situation and get through it even when things get tough.

### 4.2. Mental Health

Contrary to our second hypothesis, this study found that cancer participants were less stressed and had better eudemonic psychological wellbeing, coping better (1 in 2 flourishing), with higher positive affect and less negative affect and with higher psychological flexibility than those without cancer. The levels of stress cancer patients experienced in this study were also lower than those of frontline healthcare workers found in a recent study [77]. This interesting finding may again be related to individuals who have had a cancer experience developing the abilities to better deal with new health-related threats compared to those who had never before experienced such threat. Higher levels of stress within the group reporting cancer experiences, were associated with being female, believing that they are more susceptible to COVID-19, difficulties in obtaining basic supplies, and being less psychologically flexible. A study including a community reported females having higher levels of stress, which was confirmed in this study [78].

Closer examination of the results showed that the cancer participants’ responses to COVID-19 was not influenced by the status of active treatment as shown in other studies, including, for example, a multicenter study by Sigorski et al. [79] of 306 cancer patients of all subtypes in Poland receiving systematic therapy during the first wave of the pandemic. They showed that COVID-19 associated fear and anxiety were significantly lower than cancer-associated anxiety. Several studies on specific cancer types including sarcoma [80] and lymphoma [73] showed that patients on active treatment, particularly treatment with curative intent, were significantly more worried about both COVID-19 and cancer compared to cancer patients not on treatment (i.e., cancer survivors with their disease under control). Not surprisingly, higher anxiety and depression levels were documented in patients with advanced disease and on treatment with palliative intent versus those with curative intent [81,82]. These findings suggest that cancer patients’ mental health should be examined in light of the state of their treatment and psychological treatment provided only to those cancer patients who face difficulties.

Several studies reported increased levels of stress and anxiety in cancer patients since the start of the pandemic, with anxiety being related to their ongoing treatment, COVID-19 related hospital admissions, and severity of COVID-19 related symptoms [83,84]. Large scale epidemiological studies coming from China suggested that cancer patients tended to have overall worse mental health compared to the general population during the COVID-19 crisis [85]. In contrast, some studies [76] showed no elevation of levels of distress or COVID-19-related fear in cancer patients; instead, equally elevated COVID-19-related fear in cancer and non-cancer patients were documented in Germany. In our original study of 9565 participants, it was shown that the pandemic was experienced as a stressful situation by most people (11% reporting the highest levels of stress) [33]. Interestingly, in this study, cancer participants reported lower levels of perceived stress and depressive symptomatology compared to non-cancer participants. At the same time, they reported higher levels of positive affect, lower levels of negative affect, and a higher level of psychological flexibility compared to non-cancer participants, suggesting a flexible and positive outlook towards an unknown and otherwise frightening situation perhaps by accepting their experiences and engaging in what is important to them. Half of the cancer participants in our study were patients off cancer treatment and perhaps long-term survivors, which may partly explain the lower levels of perceived stress, higher level of acceptance, and high levels of social support and intention to help others in need. These individuals may as a result of their cancer experience have become better adapted and well-functioning, resilient, and adjustable, prepared to deal with what is otherwise a worldwide crisis.

### 4.3. Beliefs about COVID-19

This study found that cancer participants considered themselves to be more susceptible to COVID-19 and were more afraid of it than cancer at the present time (first wave). Among these cancer patients, more than 80% considered COVID-19 as their priority with no important differences on behavioral adherence to protective behaviors among those whose priority was cancer. This study showed that cancer participants—as most people across the world—felt threatened by COVID-19. Cancer participants felt they were more susceptible to being infected by COVID-19 which they considered a severe disease, more so than non-cancer participants. Not many of these participants experienced a COVID-19 infection themselves (2.3%) or had direct experience through a partner or close other (1.5% and 7.2, respectively) at the time; yet the majority (65%) reported that COVID-19 scared them the most and that COVID-19 could harm them more than anything else (71%). It could be assumed that this is related to the reports on COVID-19 during the early days of the pandemic, which suggested that the prevalence of COVID-19 in patients with cancer was high and that cancer patients comprised an at-risk population for COVID-19 [1,2]. Indeed, frequency, duration, and diversity of media exposure are related to COVID-19-asscociated fear in the general population [86] and cancer patients are likely no exception. Interestingly, however, cancer participants in this study reported shorter screen time (TV, smartphones, laptops) (Appendix A) during the study period compared to non-cancer participants; they therefore likely had less exposure to multimedia and scary information compared to non-cancer participants.

It was interesting to identify three factors consistently associated with both perceived stress and adherence to protective behaviors: the Health Belief Model parameters of perceived susceptibility and perceived severity and the Acceptance and Commitment Therapy (ACT) [87] construct of psychological flexibility. COVID-19 is a context where people’s attitudes towards the disease strongly influence how they feel and behave and add to a large body of literature on how the Health Belief Model can be used to understand health behaviors [88,89]. Psychological flexibility is comprised of inter- and intra-personal skills that refer to behaviors relating to recognizing and adapting to situations via making changes towards what is most meaningful and functional for the person [90]. There are mountains of evidence demonstrating that individuals who exhibit psychological flexibility adapt better to significant life stressors [90,91].

### 4.4. Strengths and Limitations

The limitations of the study comprise the small sample of cancer participants, variation of the study sample including the multiple national backgrounds of the participants, the different types and cancer statuses, and the varying treatments and treatment stages, participants were in. Moreover, the study did not target cancer patients only and their specific needs. It is also possible that participants’ willingness to report that they have been diagnosed with cancer may differ from country to country according to social norms and these differences may impact the generalizability of the findings. In addition, findings related to perceived susceptibility of COVID-19 may have been affected by the lack of an available COVID-19 vaccine during data collection. These parameters could alternatively be considered as strengths of the study if it was not for the relatively small number of cancer participants overall. Some of the limitations originate from the fact that this was a population-based and not a cancer-specific survey addressing issues relevant to all participants and not to cancer patients per se and perhaps their specific needs. A cross-sectional study provides a view of one time point and this study therefore reports on the mental health status at the specific time period during the first wave of the pandemic. Many of these parameters may vary with time and as such, longitudinal studies are needed to allow comparisons in time and as the pandemic progresses. Finally, this has been a nomothetical study and individual cancer patients may present with specific psychological needs that merit attention and should be considered. Thus, the good news that cancer participants managed well during the first wave of the pandemic should not lead to a dismissal of specific needs of individuals or of how their well-being may be affected in the long-term.

## 5. Conclusions

Overall, this study suggests that cancer participants were well adapted and able to handle the first wave of the pandemic and its unpredictable and uncontrollable nature. There is, however, value in proceeding to examine the cancer population on its own merit and investigate more in depth how this population has been affected, especially in the long run. As the pandemic drags on and reports of patients not being able to attend follow-up appointments and routine procedures needing to be postponed, they may bear psychological taxing effects with additional distress. All these need to be considered so that cancer patients receive adequate medical and psychological care.

## Figures and Tables

**Table 1 cancers-13-06294-t001:** Socio-demographic and medical characteristics of the participants who were recruited in the study overall and separately for cancer and non-cancer participants.

		Total (*n* = 9565)	Cancer Participants (*n* = 264, 2.8%)	Non-Cancer Participants (*n* = 9301, 97.2%)	*p*-Value	Effect Size
**Socio-demographic Characteristics**						
Age median (years (IQR)) ^a^		34 (24–46)	51.5 (40,60)	34 (26,45)	0.15 ^g^	1.05 ^i^
Gender (*n* (%)) ^b^	Male	2101 (22.0)	52 (19.7)	2049 (22.0)	0.66 ^h^	0.01 ^j^
	Female	7431 (77.7)	211 (79.9)	7220 (77.6)		
	Other/Non-Binary	33 (0.3)	1 (0.4)	32 (0.3)		
Employment status (*n* (%)) ^b^	Working full time	5108 (53.4)	134 (50.8)	4974 (53.5)	<0.01 ^h**^	0.15 ^j^
	Working part time	1674 (17.5)	49 (18.6)	1625 (17.5)		
	Unemployed	2218 (23.2)	24 (9.1)	2194 (23.6)		
	On parental leave	241 (2.2)	6 (2.3)	208 (2.2)		
	Retired	351 (3.7)	51 (19.3)	300 (2.2)		
Working as a health professional (*n* (%)) ^c^	Yes	1556 (16.6)	54 (21.3)	1502 (16.5)	0.33 ^h^	0.02 ^j^
	No	7819 (83.4)	200 (78.7)	7619 (83.5)		
University student (*n* (%)) ^d^	Yes	2718 (28.8)	26 (10.1)	2692 (29.3)	0.04 ^h*^	0.07 ^j^
	No	6735 (71.2)	232 (89.9)	6503 (70.7)		
Education level (*n* (%)) ^e^	Primary	77 (0.8)	2 (0.8)	75 (0.8)	0.74 ^h^	0.04 ^j^
	Secondary	2348 (25.2)	49 (18.6)	2299 (24.3)		
	Higher	6887 (74.0)	213 (80.6)	7097 (74.9)		
Marital status (*n* (%)) ^b^	Single	2947 (31.2)	48 (18.3)	2899 (31.6)	0.88 ^h^	0.09 ^j^
	In a relationship/Engaged/Married	5912 (62.6)	178 (68.3)	5734 (62.5)		
	Divorced/Widower	581 (6.2)	35 (13.4)	546 (5.9)		
Have children (*n* (%)) ^b^	Yes	3899 (40.8)	174 (65.9)	3725 (40.1)	0.24 ^h^	0.09 ^j^
	No	5666 (59.2)	90 (34.1)	5576 (59.9)		
Living situation (*n* (%)) ^b^	Live alone	1397 (14.6)	47 (17.8)	1350 (14.5)	0.45 ^h^	0.07 ^j^
	Live with parents	1991 (20.8)	19 (7.2)	1972 (21.2)		
	Live with one of parents	484 (5.1)	5 (1.9)	479 (5.2)		
	Live with own family	5171 (54.1)	185 (70.1)	4986 (53.6)		
	Live with friends/roommates	522 (5.5)	8 (3.0)	514 (5.5)		
**COVID-19 Infection**						
Infected by COVID-19 (*n* (%)) ^b^	Yes	135 (1.4)	6 (2.3)	129 (1.4)	0.48 ^h^	0.01 ^j^
	No	8417 (88.0)	230 (87.1)	8187 (88.0)		
	Unsure or have had symptoms but not diagnosed	1013 (10.6)	28 (10.6)	985 (10.6)		
Partner been infected by COVID-19 (*n* (%)) ^f^	Yes	70 (0.7)	4 (1.5)	66 (0.7)	0.27 ^h^	0.02 ^j^
	No	8732 (92.2)	243 (92.4)	8489 (92.2)		
	Unsure or (s)he has had symptoms but not diagnosed	670 (7.1)	16 (6.1)	654 (7.1)		
Others infected by COVID-19 (*n* (%)) ^e^	Yes	538 (5.6)	19 (7.2)	519 (5.6)	0.38 ^h^	0.01 ^j^
	No	8227 (86.0)	227 (86.0)	8000 (86.0)		
	Unsure or (s)he has had symptoms but not diagnosed	799 (8.4)	18 (6.8)	781 (8.4)		

Abbreviations: IQR, interquartile range; ^a^
*n* = 9563; ^b^
*n* = 9565; ^c^
*n* = 9375; ^d^
*n* = 9453; ^e^
*n* = 9564; ^f^
*n* = 9472; ^g^ differences between cancer participants and participants not reporting cancer were examined with *t*-test if outcomes were continuous variables with normal distributions; ^h^ differences between cancer participants and participants not reporting cancer were examined with χ^2^ test if categorical outcomes; ^i^ effect sizes were examined with Cohen’s d if outcomes were continuous variables; ^j^ effect sizes were examined with Cramér’s V if outcomes were categorical variables. * Statistically significant *p* < 0.05 ** Statistically significant *p* < 0.01.

**Table 2 cancers-13-06294-t002:** Comparisons between participants reporting cancer vs. no cancer.

		Total (*n* = 9565)	Cancer Participants (*n* = 264, 2.8%)	Non-Cancer Participants (*n* = 9301, 97.2%)	*p*-Value	Effect Size
**COVID-19 Protective Behaviors (score 0–10)**						
Keeping distance from other people when going out (median (IQR)) ^a^		9 (8, 10)	10 (9–10)	9 (8–10)	<0.01 ^c*^	0.19 ^e^
Self-isolating, limiting unnecessary travelling according to national guidelines (median (IQR)) ^a^		10 (9, 10)	10 (9–10)	10 (9–10)	0.05 ^c*^	0.03 ^e^
Washing hands regularly with water and soap (median (IQR)) ^a^		10 (9, 10)	10 (9–10)	10 (9–10)	0.04 ^c*^	0.17 ^e^
Protective behaviors self-efficacy (median (IQR))		6.2 (5.6–7)	6.6 (5.9–7)	6.2 (5.6–7)	<0.01 ^c**^	0.20 ^e^
**Coping Styles and Social Support**						
Social support (OSS)^e^ (Ν (%)) ^b^	Low	2337 (24.4)	52 (19.8)	2285 (24.6)	<0.01 ^d**^	0.03 ^f^
	Moderate	4999 (52.3)	129 (49.0)	4870 (52.4)		
	High	2227 (23.3)	82 (31.2)	2145 (23.0)		
Family functioning (BAFFS) (median (IQR))		6 (4–7)	5 (4–6)	6 (4–7)	0.45 ^c^	0.10 ^e^
**Mental Health Outcomes**						
Perceived stress (PSS) (median (IQR))		17 (12–22)	15 (10–20)	17 (12–22)	<0.01 ^c**^	0.27 ^e^
Levels of perceived stress (Ν (%)) ^b^	Low	3159 (33.0)	119 (45.3)	3040 (32.7)	<0.01 ^d**^	0.04 ^f^
	Moderate	5344 (55.9)	120 (45.6)	5224 (56.2)		
	High	1060 (11.1)	24 (9.1)	1036 (11.1)		
Psychological flexibility (Psy-Flex) (median (IQR))		34 (30, 37)	35 (32, 38)	34 (30, 37)	<0.01 ^c**^	0.37 ^e^
Mindfulness (CAMS) (median (IQR))		27 (24–29)	27 (25–30)	27 (24–29)	0.04 ^c*^	0.19 ^e^
Prosocialness (PSA) (median (IQR))		23 (20, 26)	23 (20–26)	23 (20–26)	0.99 ^c^	0.01 ^e^
Depressive symptomatology (MSBS—reinforcement) (median (IQR))		3 (3–3)	3 (3–3)	3 (2,3)	0.77 ^c^	0.10 ^e^
Depressive symptomatology (MSBS -boredom) (median (IQR))		2 (1.5–3)	2 (1.5–2.5)	2 (1.5–3)	<0.01 ^c**^	0.24 ^e^
Wellbeing total (MHCSF) (median (IQR))		42 (31–52)	46 (36–54)	42 (31–52)	<0.01 ^c**^	0.22 ^e^
Hedonic wellbeing (MHCSF) (median (IQR))		11 (8–12)	12 (9–13)	11 (8–12)	<0.01 ^c**^	0.19 ^e^
Eudemonic social wellbeing (MHCSF) (median (IQR))		11 (7–16)	12 (8–17)	11 (7–16)	0.21 ^c^	0.16 ^e^
Eudemonic psychological wellbeing (MHCSF) (median (IQR))		21 (15–24)	23 (18–26)	21 (15–24)	<0.01 ^c**^	0.24 ^e^
Wellbeing type (MHCSF)	Languishing	882 (10.1)	20 (8.1)	862 (10.2)	<0.01 ^d**^	0.04 ^f^
	Moderately mentally healthy	4345 (50.0)	104 (41.9)	4241 (50.2)		
	Flourishing	3468 (39.9)	124 (50.0)	3344 (39.6)		
Positive affect (PANAS) (median (IQR))		29 (23–35)	30 (25–37)	29 (23–35)	0.01 ^c*^	0.21 ^e^
Negative affect (PANAS) (median (IQR))		28 (21–38)	26 (20–33)	28 (21–38)	0.01 ^c*^	0.19 ^e^
**Beliefs about COVID-19**						
Perceived susceptibility (median (IQR))		9 (6–11)	11 (7.5, 14)	8 (6–11)	<0.01 ^c**^	0.56 ^e^
Perceived severity (median (IQR))		13 (10–15)	14 (12–17)	13 (10–15)	<0.01 ^c**^	0.37 ^e^

Abbreviations: BAFFS = Brief Assessment for Family Functioning Scale; CAMS = Cognitive and Affective Mindfulness Scale; IQR = interquartile range; MHCSF = Mental Health Continuum Short Form; MSBS = Multidimensional State Boredom Scale; OSS = Oslo Social Supports Scale; PANAS = Positive and Negative Affect Scale; PSA = Prosocialness Scale for Adults; PSS = Perceived Stress Scale; ^a^ Ν = 9565; ^b^ N = 9563 ^c^ differences between cancer and non-cancer participants were examined with χ^2^ test; ^d^ differences between cancer and non-cancer participants were examined with Kolmogorov–Smirnov test; ^e^ effect size between cancer and non-cancer participants was examined with Cohen’s d; ^f^ effect size between cancer and non-cancer participants was examined with Cramér’s V. * Statistically significant *p* < 0.05 ** Statistically significant *p* < 0.01.

**Table 3 cancers-13-06294-t003:** Multivariate linear regression of the level of keeping distance as a linear combination of social support (model 1), perceived susceptibility and perceived severity (model 2), change in finances during quarantine and availability of obtaining all the basic supplies (model 3), psychological flexibility (model 4), and all the predictors (model 5) after adjusting for different socio-demographic and socio-economic characteristics (i.e., age, gender).

	Coefficient				
	Model 1	Model 2	Model 3	Model 4	Model 5
**Priority of Health Concern during Pandemic**					
Cancer	Ref	Ref	Ref	Ref	Ref
COVID-19	0.57 (0.07, 1.07) *	0.29 (−0.23, 0.80)	0.63 (0.13, 1.14) *	0.62 (0.12, 1.11) *	0.43 (−0.09, 0.94)
**Age**	0.01 (−0.01, 0.02)	0.00 (−0.02, 0.02)	0.01 (−0.01, 0.02)	0.00 (−0.02, 0.02)	−0.01 (−0.02, 0.01)
**Gender**					
Male	Ref	Ref	Ref	Ref	Ref
Female	−0.11 (−0.78, 0.56)	−0.17 (−0.87, 0.46)	−0.02 (−0.71, 0.67)	−0.18 (−0.85, 0.48)	−0.18 (−0.86, 0.49)
**Cancer Type**					
Thyroid	Ref	Ref	Ref	Ref	Ref
GI	0.31 (−0.71, 1.33)	0.11 (−0.89, 1.10)	0.42 (−0.60, 1.44)	0.41 (−0.59, 1.42)	0.33 (−0.66, 1.32)
Breast	0.33 (−0.45, 1.11)	0.38 (−0.41, 1.10)	0.43 (−0.35, 1.22)	0.38 (−0.38, 1.15)	0.48 (−0.27, 1.23)
Soft molecular	−0.84 (−3.97, 2.29)	−0.56 (−3.51, 2.57)	−0.83 (−3.95, 2.29)	−0.69 (−3.76, 2.39)	−0.28 (−3.28, 2.72)
Female reproductive	0.43 (−0.47, 1.33)	0.34 (−0.54, 1.21)	0.56 (−0.35, 1.47)	0.43 (−0.45, 1.32)	0.49 (−0.38, 1.36)
Leukemia	0.29 (−0.72, 1.30)	0.30 (−0.67, 1.29)	0.43 (−0.60, 1.46)	0.26 (−0.73, 1.25)	0.43 (−0.56, 1.42)
Lymphoma	0.19 (−0.85, 1.24)	0.08 (−0.98, 1.05)	0.25 (−0.79, 1.29)	0.32 (−0.71, 1.34)	0.16 (−0.84, 1.17)
Cervical	0.01 (−1.70, 1.71)	−0.20 (−1.89, 1.42)	0.14 (−1.56, 1.85)	−0.09 (−1.77, 1.59)	−0.18 (−1.82, 1.47)
Skin	0.25 (−0.69, 1.18)	0.24 (−0.67, 1.14)	0.34 (−0.59, 1.28)	0.22 (−0.69, 1.14)	0.32 (−0.58, 1.21)
Lung	1.06 (−0.88, 3.00)	1.35 (−0.70, 3.06)	1.24 (−0.69, 3.16)	1.18 (−0.71, 3.07)	1.28 (−0.58, 3.13)
Urine	0.34 (−0.99, 1.66)	0.23 (−0.97, 1.59)	0.37 (−0.96, 1.69)	0.50 (−0.80, 1.80)	0.57 (−0.71, 1.85)
Testicular	0.53 (−1.49, 2.54)	0.33 (−1.62, 2.32)	0.52 (−1.49, 2.53)	0.04 (−1.97, 2.04)	−0.01 (−1.96, 1.96)
Prostate	0.57 (−0.83, 1.98)	0.87 (−0.59, 2.15)	0.82 (−0.60, 2.24)	0.54 (−0.83, 1.91)	0.92 (−0.46, 2.30)
Bone	0.22 (−1.45, 1.90)	0.50 (−1.18, 2.08)	0.26 (−1.41, 1.92)	0.16 (−1.48, 1.80)	0.34 (−1.26, 1.95)
Brain	0.13 (−1.53, 1.80)	0.42 (−1.13, 2.12)	0.10 (−1.56, 1.76)	0.17 (−1.46, 1.81)	0.54 (−1.06, 2.14)
**Cancer Therapy**					
Yes	Ref	Ref	Ref	Ref	Ref
No	−0.39 (−0.90, 0.12)	−0.31 (−0.81, 0.19)	−0.46 (−0.97, 0.05)	−0.39 (−0.89, 0.10)	−0.38 (−0.88, 0.11)
**Social Support**	0.05 (−0.04, 0.14)				0.03 (−0.06, 0.12)
**Beliefs about COVID-19**					
Perceived susceptibility		−0.08 (−0.14, −0.01) *			−0.07 (−0.13, −0.01) *
Perceived severity		0.15 (0.07, 0.23) *			0.14 (0.06, 0.22) *
**Finances in Quarantine**					
Have got better			Ref		Ref
Stayed the same			1.12 (0.03, 2.21) *		1.18 (0.13, 2.23) *
Have got worse			1.25 (0.13, 2.38) *		1.36 (0.27, 2.45) *
**Access to Basic Supplies**					
Yes			Ref		Ref
No			−0.20 (−1.08, 0.67)		−0.13 (−0.97, 0.72)
**Psychological Flexibility**				0.06 (0.02, 0.10) *	0.05 (0.01, 0.09) *

Note: * Statistically significant *p* < 0.05.

**Table 4 cancers-13-06294-t004:** Multivariate linear regression of stress as a linear combination of social support (model 1), perceived susceptibility and perceived severity (model 2), change in finances during quarantine and availability of obtaining all the basic supplies (model 3), psychological flexibility (model 4), and all the predictors (model 5) after adjusting for different socio-demographic and socio-economic characteristics (i.e., age, gender).

	Coefficient (*p*-Value)				
	Model 1	Model 2	Model 3	Model 4	Model 5
**Priority of Health Concern during Pandemic**					
Cancer	Ref	Ref	Ref	Ref	Ref
COVID-19	0.73 (−1.50, 2.97)	0.66 (−1.69, 2.92)	0.97 (−1.22, 3.17)	0.38 (−1.71, 2.46)	0.29 (−1.82, 2.41)
**Age**	−0.16 (−0.24, −0.08) *	−0.12 (−1.20, −0.04) *	−0.15 (−0.23, −0.08) *	−0.10 (−0.18, −0.03) *	−0.07 (−0.14, 0.01)
**Gender**					
Male	Ref	Ref	Ref	Ref	Ref
Female	0.53 (−2.46, 3.52)	1.49 (−1.51, 4.50)	1.41 (−1.59, 4.41)	1.16 (−1.63, 3.95)	3.01 (0.24, 5.78) *
**Cancer Type**					
Thyroid	Ref	Ref	Ref	Ref	Ref
GI	−1.03 (−5.57, 3.50)	−1.27 (−5.77, 3.22)	−0.33 (−4.79, 4.12)	−2.00 (−6.23, 2.22) *	−1.48 (−5.56, 2.60) *
Breast	−3.99 (−7.45, −0.53) *	−4.40 (−7.81, −1.00) *	−3.80 (−7.20, −0.40) *	−4.43 (−7.65, −1.21) *	−4.32 (−7.40, −1.24) *
Soft molecular	−13.12 (−27.03, 0.79)	−13.99 (−27.70, −0.27) *	−11.05 (−24.64, 2.54)	−14.45 (−27.40, −1.50) *	−14.21 (−26.54, −1.88) *
Female reproductive	−4.70 (−8.68, −0.67) *	−4.67 (−8.62, −0.72) *	−4.47 (−8.41, −0.52) *	−4.71 (−8.43, −0.98) *	−4.68 (−8.25, −1.10) *
Leukemia	−5.60 (−10.07, −1.12) *	−6.25 (−10.66, −1.83) *	−5.95 (−10.41, −1.48) *	−5.35 (−9.52, −1.19) *	−6.39 (−10.45, −2.33) *
Lymphoma	−7.52 (−12.16, −2.90) *	−7.85 (−12.42, −3.28) *	−7.56 (−12.10, −3.03) *	−8.76 (−13.07, 4.45)	−8.11 (−12.25, −3.98) *
Cervical	−8.10 (−15.67, −0.52) *	−7.40 (−14.87, 0.07)	−9.03 (−16.47, −1.59) *	−7.21 (−14.27, −0.16) *	−7.46 (−14.21, −0.70) *
Skin	−6.62 (−10.77, −2.48) *	−6.40 (−10.47, −2.32) *	−6.65 (−10.71, −2.59) *	−6.43 (−10.29, −2.57) *	−6.36 (−10.04, −2.68) *
Lung	−5.95 (−14.56, 2.66)	−7.70 (−16.12, 0.73)	−8.08 (−16.46, 0.30)	−7.19 (−15.15, 0.77)	−7.72 (−15.36, −0.09) *
Urine	−0.88 (−6.74, 4.98)	−0.05 (−5.80, 5.69)	1.37 (−4.38, 7.12)	−2.26 (−7.73, 3.21)	−0.69 (−5.94, 4.57)
Testicular	−13.54 (−22.49, −4.59) *	−10.95 (−19.84, −2.05) *	−13.09 (−21.83, −4.35) *	−9.10 (−17.50, −0.62) *	−7.83 (−15.90, 0.24)
Prostate	−6.16 (−12.38, 0.07)	−5.97 (−12.15, 0.20)	−5.10 (−11.29, 1.09)	−5.90 (−11.68, −0.12) *	−3.64 (−9.30, 2.03)
Bone	1.36 (−6.06, 8.79)	−0.21 (−7.54, 7.11)	0.47 (−6.79, 7.73)	1.91 (−5.01, 8.82)	0.63 (−5.97, 7.23)
Brain	0.53 (−6.87, 7.93)	0.17 (−7.15, 7.48)	1.99 (−5.24, 9.22)	0.25 (−6.64, 7.13)	0.27 (−6.32, 6.86)
**Cancer Therapy**					
Yes	Ref	Ref	Ref	Ref	Ref
No	−1.00 (−3.26, 1.25)	−0.49 (−2.75, 1.77)	−1.05 (−3.27, 1.17)	−0.94 (−3.05, 1.16)	−0.58 (−2.62, 1.46)
**Social Support**	−0.57 (−0.96, −0.17) *				−0.18 (−0.56, 0.19)
**Beliefs about COVID-19**					
Perceived susceptibility		0.57 (0.28, 0.86) *			0.42 (0.15, 0.68) *
Perceived severity		−0.26 (−0.62, 0.10)			−0.18 (−0.51, 0.15)
**Finances in Quarantine**					
Have got better			Ref		Ref
Stayed the same			−0.18 (−4.93, 4.56)		−1.40 (−5.70, 2.91)
Have got worse			3.49 (−1.41, 8.38)		1.73 (−2.73, 6.20)
**Access to Basic Supplies**					
Yes			Ref		Ref
No			5.69 (1.88, 9.50) *		4.97 (1.49, 8.44) *
**Psychological Flexibility**				−0.54 (−0.71, −0.38) *	−0.42 (−0.60, −0.25) *

Note: * Statistically significant *p* < 0.05.

**Table 5 cancers-13-06294-t005:** Comparisons between cancer participants who had cancer as their priority vs. those who had COVID-19 as their priority.

		Cancer Priority (*n* = 52)	COVID-19 Priority (*n* = 191)	*p*-Value	Effect Size
**Socio-demographic Characteristics**					
Median age of participants (years (IQR)) ^a^		54.5 (44, 57.5)	50 (39, 60)	0.15 ^m^	0.07 ^n^
Gender (*n* (%)) ^a^	Males	12 (23.1)	35 (18.3)	0.66 ^k^	0.06 ^o^
	Females	40 (76.9)	155 (81.2)		
	Other/Non-Binary	0 (0.0)	1 (0.5)		
Employment status (*n* (%)) ^a^	Working full time	134 (50.8)	4974 (53.5)	<0.01 ^k**^	0.15 ^o^
	Working part time	49 (18.6)	1625 (17.5)		
	Unemployed	24 (9.1)	2194 (23.6)		
	On parental leave	6 (2.3)	208 (2.2)		
	Retired	51 (19.3)	300 (2.2)		
Working as a health professional (*n* (%)) ^b^	Yes	13 (26.0)	36 (19.7)	0.33 ^k^	0.06 ^o^
	No	37 (74.0)	147 (80.3)		
University student (*n* (%)) ^c^	Yes	1 (2.0)	22 (11.8)	0.04 ^k*^	0.14 ^o^
	No	50 (98.0)	164 (88.2)		
Education level (*n* (%)) ^d^	Primary	0 (0.0)	2 (1.1)	0.74 ^k^	0.09 ^o^
	Secondary	9 (17.3)	34 (18.2)		
	Higher	43 (82.7)	151 (80.7)		
Marital status (*n* (%)) ^e^	Single	10 (19.6)	34 (18.0)	0.88 ^k^	0.10 ^o^
	In a relationship/Engaged/Married	35 (68.6)	128 (67.7)		
	Divorced/Widower	6 (11.8)	27 (14.3)		
Having children (*n* (%)) ^a^	Yes	37 (71.2)	119 (62.3)	0.24 ^k^	0.08 ^o^
	No	15 (28.8)	72 (37.7)		
Living situation (*n* (%)) ^a^	Living alone	7 (13.5)	38 (19.9)	0.45 ^k^	0.12 ^o^
	Living with parents	5 (9.6)	14 (7.3)		
	Living with one of parents	2 (3.9)	3 (1.6)		
	Living with own family	38 (73.1)	131 (68.6)		
	Living with friends/roommates	0 (0.0)	5 (2.6)		
Type of cancer (*n* (%)) ^f^	Breast cancer	19 (36.5)	47 (25.7)	0.67	0.23 ^o^
	Female reproductive system cancers	3 (5.8)	24 (13.1)		
	Cervical cancers	0 (0.0)	4 (22)		
	Thyroid cancer	4 (7.7)	19 (10.4)		
	GI cancer	7 (13.5)	10(5.5)		
	Leukemia and blood cancers	3 (5.8)	14 (7.7)		
	Lymphomas	4 (7.7)	13 (7.1)		
	Testicular cancers	0 (0.0)	3 (1.6)		
	Lung cancers	3 (5.8)	5 (2.7)		
	Skin cancers	4 (7.7)	21 (11.5)		
	Brain cancers	1 (1.9)	3 (1.6)		
	Prostate cancers	2 (3.8)	8 (4.4)		
	Bone cancers	0 (0.0)	4 (2.2)		
	Urine cancers	2 (3.8)	6 (3.3)		
	Soft molecular cancers	0 (0.0)	2 (1.1)		
In therapy (*n* (%)) ^d^	Yes	15 (28.8)	40 (20.9)	0.23 ^k^	0.08 ^o^
	No	37 (71.2)	151 (79.1)		
Type of therapy (*n* (%)) ^g^	Adjuvant preventative	12 (70.6)	31 (73.8)	0.80 ^k^	0.20 ^o^
	For active disease	5 (29.4)	11 (26.2)		
The scariest thing at this moment? (*n* (%)) ^h^	Cancer	44 (89.8)	37 (19.9)	<0.01 ^k**^	0.60 ^o^
	COVID-19	5 (10.2)	149 (80.1)		
The most harmful thing at this moment? (*n* (%)) ^i^	Cancer	43 (84.3)	28 (14.8)	<0.01 ^k**^	0.62 ^o^
	COVID-19	8 (15.7)	161 (85.2)		
**COVID-19 Protective Behaviors**					
Keeping distance from other people when going out (median (IQR)) ^a^		10 (8.5–10)	10 (9–10)	0.77 ^l^	0.28 ^n^
Self-isolating, limiting unnecessary travelling according to national guidelines (median (IQR)) ^a^		10 (9–10)	10 (9–10)	0.16 ^l^	0.38 ^n^
Washing hands regularly with water and soap (median (IQR)) ^a^		10 (9–10)	10 (9–10)	0.99 ^l^	0.20 ^n^
**Coping and Social Support**					
Social support (OSS) ^j^ (*n* (%)) ^h^	Low	9 (17.3)	43 (22.6)	0.70 ^k^	0.05 ^o^
	Moderate	25 (48.1)	88 (46.3)		
	High	18 (34.6)	59 (31.1)		
**Perceived Stress (PSS)**					
Score^e^ (*n* (%)) ^h^	Low	24 (46.1)	81 (42.6)	0.58 ^k^	0.07 ^o^
	Moderate	25 (48.1)	89 (46.8)		
	High	3 (5.8)	20 (10.6)		
**Beliefs about COVID-19**					
Perceived susceptibility (median (IQR))		9.5 (6.5–13)	11 (8–14)	0.34 ^g^	0.28 ^n^
Perceived severity (median (IQR))		12.5 (9–15)	15 (13–17)	<0.01 ^g**^	0.74 ^n^

Abbreviations: SD = standard deviation; IQR = interquartile range; OSS = Oslo Social Support Scale; PSS = Perceived Stress Scale ^a^
*n* = 264; ^b^
*n* = 254; ^c^
*n* = 258; ^d^
*n* = 260; ^e^
*n* = 261; ^f^
*n* = 235; ^g^
*n* = 145; ^h^
*n* = 242; ^i^
*n* = 244; ^j^
*n* = 263; ^k^ differences between cancer priority and COVID priority were evaluated by the chi-square test. ^l^ Differences between cancer priority and COVID priority were evaluated by the Kolmogorov–Smirnov test. ^m^ differences between cancer priority and COVID priority were evaluated by the *t*-test; ^n^ effect size between males and females was examined with Cohen’s d; ^o^ effect size between males and females was examined with Cramér’s V. * Statistically significant *p* < 0.05 ** Statistically significant *p* < 0.01.

## Data Availability

The data from this study are available by the corresponding author on reasonable request.

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
