# Peer review of "Mental Health and Adherence to COVID-19 Protective Behaviors among Cancer Patients during the COVID-19 Pandemic: An International, Multinational Cross-Sectional Study"

_cancers, 2021, doi:10.3390/cancers13246294_

Round 1
Reviewer 1 Report
Thank you for the opportunity to review the manuscript entitled 'Mental health and adherence to COVID-19 protective behaviors among cancer patients during the COVID-19 pandemic: An international, multinational cross-sectional study'.
In my opinion the proposed manuscript does not fulfill the high standards for publications of Cancers. There are several reasons that have led me to my decision to recommend rejection of the manuscript. I presented some of them below.
- This topic is not new and the data is comparable old, there is no added value for the community
- There a several studies the authors did not mentioned in the introduction (cancer/ risk-group related studies on mental health during the pandmic)
- This study was not planed to investigate cancer patients. The data-driven selection of cancer patients leads to several biases!
- The authors only named objectives, but no hypotheses
- Very small sample size for this research design (multi-country cross sectional study)
- "Written informed consent" was definitive not obtained before participation. The authors should state this! Probably electronic informed consent was obtained.
- Supplement Material does not cover all of the named aspects in the text
- Validity of the outcomes is not clear
-The limitation section does not cover all relevant aspects (see for some of them above)
Reviewer 2 Report
On page 3, the authors argued that little is known about the impact of COVID-19 on cancer patients’ mental health and protective behaviors. Given that the manuscript compared cancer patients with noncancer patients on their adherence to protective behaviors and mental health, the authors may want to elaborate on the expected differences.
Page 3 (line 120-128): the flow of the paragraph is not smooth. It is better to focus on the aim (lines 118-120). The link between mental health and psychological distress, anxiety and so on is missing and the discussion of “psychological flexibility and mindfulness qualities” (line 127) seems to come from nowhere. A proper link between those concepts would improve the logical flow of the manuscript. Btw, “responsible for certain behaviors” (line 126) is too vague and the elaboration on adherence is missing.
Even though this study is part of a larger survey, it is better to provide more information about the participants and data collection procedure (e.g., consent, any incentive for participation, ethics approval) because readers may not be familiar with other research outputs coming from this data set.
Would it be possible that the willingness to report having or not having cancer in the survey affected by countries? Would these differences, if any, affect the interpretations and the generalizability of the findings?
No vaccine is available at the time of data collection. Authors may want to address if it may have an impact on the findings, and if applicable, as a limitation of the study.
Reviewer 3 Report
Overall comments
- The study is a multinational study comprising a large population. The study has used multiple previously validated tools to assess the impact and behavior of participants. However, a write-up needs to be revisited where required as mentioned below.
- The status of COVID 19 in the nation could also have impacted the perception and behavior of cancer diagnosed or normal residents of the nation. Regardless of those factors, the study has included multiple validated tools and associated variables to understand the impact and behavior. However, at some areas, authors need attention as listed below
Specific comments
- The introduction section is too lengthy and the second the last paragraph seems to be part of the methods “In this context, through an initiative led by members of our group, an international population based cross-sectional study was conducted during the first lockdown period (April 2020 – June 2020) to explore how people across the world reacted to the COVID-19 by examining outcomes of stress, depression, affect, and wellbeing by country of residence, social demographic characteristics, COVID-19 lockdown related predictors, social and psychological predictors, using an anonymous online survey [27]. With the participation of 9565 people from 78 countries, the results showed that on average about 10% of the sample had low levels of mental health whilst 50% had only moderate mental health with three consistent predictors of mental health wellbeing: perceived social support, education level, and psychologically flexible (vs. rigid) responding. Poorer outcomes were most strongly predicted by worsening of finances and no access to basic supplies.”
- Line 158: clarify the phrase “any country in the world”, originally from any nation or residing in any nation
- Line 164: Please clarify the total number of participating universities in the total (? 78) nations.
- In methods, it is not mentioned how single participants enrolled for a single time only and how the repetition of participants was omitted?
- Though the introduction was written with a large chunk of references, the discussion comparison is less robust and the authors mainly focused on cancer patients' findings and did not contrast much with the general population or other groups’ mental health. So authors are advised to compare cancer patients’ findings with other general populations. May check the following references and can include other similar pertinent references from different parts of the globe like developed versus developing nations, etc. Pertinent references for such discussion can be
https://pubmed.ncbi.nlm.nih.gov/33023563/, https://pubmed.ncbi.nlm.nih.gov/34345537/
- Line 224 225: Please clarify the statement or rephrase it to better present which two variables were compared with each other
Round 2
Reviewer 1 Report
I think that the revisions have contributed to an improvement in the quality of the manuscript, but I still consider the methodological difficulties insurmountable. I recommend submission to another journal.